# Does PLEX^®^ Elite 9000 OCT Identify and Characterize Most Posterior Pole Lesions in Highly Myopic Patients?

**DOI:** 10.3390/jcm12051846

**Published:** 2023-02-25

**Authors:** Pablo Arlanzon-Lope, Miguel Angel. Campos, Ivan Fernandez-Bueno, Rosa M. Coco-Martin

**Affiliations:** Retina Group, Instituto de Oftalmobiologia Aplicada (IOBA), Universidad de Valladolid, 47011 Valladolid, Spain

**Keywords:** pathologic myopia, OCT, Plex Elite 9000

## Abstract

High myopia (HM) is defined as an axial length (AL) ≥ 26 mm that may result in various pathologies that constitute pathologic myopia (PM). The PLEX^®^ Elite 9000 (Carl Zeiss AC, Jena, Germany) is a new swept-source optical coherence tomography (SS-OCT) underdevelopment that allows wider, deeper and more detailed posterior-segment visualization; it can acquire ultra-wide OCT angiography (OCTA) or new ultra-wide high-density scans in one image. We assessed the technology’s ability to identify/characterize/quantify staphylomas and posterior pole lesions or image biomarkers in highly myopic Spanish patients and estimate the technology’s potential to detect macular pathology. The instrument acquired 6 × 6 OCTA, 12 × 12 or 6 × 6 OCT cubes, and at least two high-definition spotlight single scans. A hundred consecutive patients (179 eyes; age, 51.4 ± 16.8 years; AL, 28.8 ± 2.33 mm) were recruited in one center for this prospective observational study. Six eyes were excluded because images were not acquired. The most common alterations were perforating scleral vessels (88.8%), classifiable staphyloma (68.7%), vascular folds (43%), extrafoveal retinoschisis (24%), dome-shaped macula (15.6%), and more uncommonly, scleral dehiscence (4.46%), intrachoroidal cavitation (3.35%), and macular pit (2.2%). The retinal thickness of these patients decreased, and the foveal avascular zone increased in the superficial plexus compared with normal eyes. SS-OCT is a novel potent tool that can detect most main posterior pole complications in PM and may provide us with a better understanding of the associated pathologies; some pathologies were identifiable only with this new kind of equipment, such as perforating scleral vessels, which seem to be the most common finding and not so frequently related to choroidal neovascularization, as previously reported.

## 1. Introduction

According to the International Myopia Institute (IMI), any refractive defect whose spherical equivalent is −0.50 diopter (D) or less is considered to be myopia, while high myopia (HM) is defined as a refractive error of −6 D or less or an axial length (AL) exceeding 26 mm [1,2]. However, pathologic myopia (PM) is defined as an axial HM with either staphyloma or myopic maculopathy (MM), the latter of which includes myopic macular degeneration (MMD), tractional maculopathy (TM), or dome-shaped macula (DSM) [3,4].

Staphyloma was defined by Spaide as an outpouching of the ocular globe wall in which the curvature radius is smaller compared to that of the surrounding area [5]. In 1977, Curtin defined 10 types of staphylomas using funduscopy to show the one that affects the posterior pole and the inferior lesion type was the most common [6]. Ohno-Matsui later proposed six types of staphylomas using fundus photographs and three-dimensional magnetic resonance [7]. Eyes with staphyloma have a high risk of having MMD, which would include diffuse or patchy macular atrophy with/without lacquer cracks that can be observed via optical coherence tomography (OCT) imaging as defects in Bruch’s membrane and active choroidal neovascularization (CNV) and subsequent Fuch’s spots [8].

Several authors have proposed classifications of MM. First, Curtin and Karlin proposed a classification including the following categories: chorioretinal atrophy, Fuch’s spot, optic nerve damage, lacquer cracks, and posterior staphyloma [9]. Later, Avila et al. [10] proposed a classification using five levels. Ohno-Matsui et al. [11] then proposed the international classification and grading system for MM called META-PM, which has been widely used and is based on retinographies. This classification has the following categories: 0 corresponds to a normal fundus appearance, 1 to tessellated fundus, 2 to diffuse atrophy, 3 to patchy atrophy without foveal involvement, and 4 to macular atrophy. The presence of any of these three plus lesions (lacquer cracks, CNV, and Fuch’s spot) also indicates PM [11]. However, this classification does not consider the tractional component of MM, which is an important cause of visual loss in PM [12]; thus, Ruiz-Medrano et al. proposed the ATN classification using both fundus and OCT images where A indicates atrophic, T tractional, and N neovascular classifications. There are different levels for each component and the final classification is the combination of the three [4]. Finally, a DSM or the presence of a ridge-shaped macula (RSM) in cases of inferior staphylomas has also been included in the IMI OCT classification as a feature of MM [3], with RSM defined as macular elevation in only one meridian across the fovea and differing from DSM, which is an area of elevation [13].

Structural OCT is a useful method to study PM features and the best method to identify tractional maculopathy [14]. In addition, one of the most important causes of visual loss in PM is CNV [15,16], and recently, OCT angiography (OCTA) has shown good sensitivity for detecting it [17,18]. Thus, structural OCT and OCTA are important for detecting macular pathologies that are otherwise difficult to identify.

It is difficult to estimate the prevalence of PM/MM and the related lesions because large population studies with large patient samples are needed. A recent review found that studies that use different definitions of MM and different diagnostic techniques are mainly conducted in Asian populations, and the prevalence of HM varied from 1.5% to 8% and the prevalence of MM from 0.2% to 10.7% [19]. However, two population studies in Germany and Russia showed prevalence rates of MM of 0.5% and 1.3%, respectively. The prevalence of HM is high in Spain and one of the highest in Europe, although it is even higher in Asian populations [20]. A study published in 2010 found that MP was the second most common cause of visual impairment in a nursing home [21]. In addition, our previous study showed differences between our population and Asian populations [22].

Concerning studies that use OCT, Ruão et al. found that patchy retinal atrophy is the most frequent finding (79.1%), followed by CNV (69.1%) and TM (53.7%) in a hospital sample of patients with HM from the Iberian Peninsula [23]. However, it is necessary to acknowledge that HM is associated with reduced signal strength spectral-domain OCT and it is more likely to produce unreliable OCT measurements [24].

The purpose of the current study was to assess whether wide-field SS-OCT and OCTA images reinforced by funduscopy/retinography allow the identification and characterization of staphyloma and the most posterior pole lesions or image biomarkers in a series of Spanish patients with HM. We also wanted to estimate the frequency of observations of the previously described pathological features and study the potential high-definition swept-source OCT (SS-OCT), which is still being developed, for detecting macular pathologies.

## 2. Materials and Methods

This longitudinal observational study of a single-center, prospective cohort followed the tenets of the Helsinki Declaration of 1964 (last amendment, 2013). The Clinical Research Ethics Committee of the Valladolid East Health Area approved the study. All patients understood the specifics of the study and provided informed consent. 

A sample of consecutive patients with HM was recruited at the Institute of Applied Ophthalmobiology (IOBA), an eye institute of the University of Valladolid. Patients were included if they were older than 18 years, Caucasian, and had an AL ≥ 26 mm. Patients were excluded if they had any other retinal pathology not attributable to HM, significant media opacities that prevented good-quality images, or those who had undergone surgery during the previous 3 months.

Family and past medical antecedents were gathered, including previous ocular pathologies and surgeries. Visual acuity (VA) was tested using an Early Treatment Diabetic Retinopathy Study panel and recorded as the logarithmic of the minimum angle of resolution (logMAR) scale. A phenylephrine (100 mg/mL) and a tropicamide (10 mg/mL drop) was instilled in each eye. The AL was measured using the IOLMaster 500 (Carl Zeiss AC, Jena, Germany). Keratometry, anterior chamber depth, and white-to-white distance were also recorded as control variables. Central retinography was later performed using either Topcon 3D (Topcon Corporation, Tokyo, Japan) or Topcon Triton DRI (Topcon Corporation). Finally, several protocol scans using the PLEX^®^ Elite 9000 OCT (Carl Zeiss AC) were performed. These included a 6 × 6 OCTA scan centered on the macula and a 12 × 12 HD51 scan centered on the macula; however, if the quality precluded it, a 6 × 6 HD51 scan was performed instead. A total of 120 6 × 6 HD51 scans and 59 12 × 12 scans were taken. Finally, at least 2 single HD spotlight line scans (90 and 180 degrees, 16 mm) were performed. To better characterize the type of staphyloma in some patients, extra scans were taken at 135° in their right eyes and 45° in their left eyes. All tests were performed by two equally trained operators (PAL and MAC). 

The presence of staphyloma was assessed using OCT. The staphylomas were classified in a similar manner to the Ohno-Matsui criterion [7]. In addition, the following lesions were identified using OCT: lacquer cracks were defined as linear Bruch defects with OCT and retinography; atrophic maculopathy was defined as extensive Bruch’s defect with OCT; if there were CNV or Fuch´s spot they were usually visible with OCTA, where active CNV may associate hemorrhage and/or exudation signs, whereas in its scar phase flow is less likely to be detected and that has the typical blackened appearance of Fuch’s spots on retinography [3,4]. Tractional maculopathy is defined by the T dominion of the ATN classification that on OCT images is shown as a split between the inner and outer retina within the staphyloma defined as foveoschisis or extrafoveal retinoschisis, which may be associated with lamellar holes or epiretinal membranes; however, preretinal membranes are often hard to distinguish from inner laminar membrane detachment in high myopia [3,4]. DSM/RSM is recognized as an inward bulge of the macula within the chorioretinal posterior concavity of the eye at the macular location in OCT that is sometimes associated with serous macular detachment [3]. Presence of peripapillary atrophy was subclassified by us as (1) myopic conus, (2) circumpapillary atrophy, (3) circumpapillary atrophy involving the macula, or (4) extensive circumpapillary atrophy involving the fovea. A tilted disk was considered to be present if the optic nerve entered the eye at an oblique angle, usually infero-nasal, while being rotated along its anterior–posterior axis [25]. We also looked for rare findings, such as macular pits defined as a sharp retina, RPE, choroid and scleral outpouching usually located within an area of patchy macular atrophy; choroidal cavitation defined as an absence of the choroidal space that is preferably located adjacent to the optic disc; scleral dehiscence defined as full-thickness scleral defects, together with outward displacement of the retina; vascular folds defined as paravascular inner retinal cleavage due to the traction of the vessels within a staphyloma detected on OCT; and perforating scleral vessels (PSV) defined as hypofluorescent vessel-like areas breaking through the sclera and reaching the choroid, demonstrating the relationship between them and other macular lesions [26,27,28]. The foveal avascular zone (FAZ) area and its perimeter at the superficial and deep retinal plexus were then manually measured using the caliper tool in the 6 × 6 OCTA macula-centered scans, where the segmentation was automatically performed by the software. The retinal thickness was measured manually from the external limiting membrane to the Bruch–retinal pigmented epithelium (B-RPE) complex. The choroidal thickness was measured from the B-RPE complex to the choroidal–scleral junction. The scleral thickness was measured from the choroidal–scleral junction to the end of the visible sclera. The vertical HD spotlight scan that produced the measurements was performed by one researcher (PAL). The presence of the cilioretinal artery was evaluated using retinography and classified according to the criterion of Meng et al. [29]. 

The appropriate sample size was calculated for estimating the proportion of staphyloma that was considered as the principal variable. A sample size of 100 eyes was obtained to estimate an expected proportion of 0.5 with an accuracy of 0.1 and a statistical power and significance level of 80% and 0.05, respectively.

Data were collected in an Excel sheet (MS, Redmond, WA, USA) and statistical analysis was performed using the Statistical Package for the Social Sciences version 23 (IBM, Armonk, NY, USA) for Windows. Quantitative variables are expressed as the mean ± standard deviation (SD) and categorical variables as percentages.

## 3. Results

A total of 179 eyes of 100 patients were studied (68.7% women; age, 50.58 ± 14.98 years; VA, 0.18 ± 0.29 logMAR; AL, 28.83 ± 2.34 mm). A total of 6 of the 21 excluded eyes were rejected due to poor-quality OCT images, and the rest did not fulfil the inclusion criteria (mainly second eyes with AL < 26 mm of anisometropic patients). Of the six eyes excluded for poor images, the reasons for exclusion were poor fixation for very low VA (four eyes), intraocular lens opacification (one eye), and endotropia (one eye).

Staphyloma was detected in the OCT images in 123 (68.7%) eyes, and it was possible to classify all of them. The types of staphylomas detected and classified based on OCT imaging are presented in Figure 1. 

The most common types of staphyloma were the wide and the narrow macular types, followed by the nasal and peripapillary types and the least common were the inferior and other types. The images of each type of staphyloma are presented in Figure 2.

Atrophic maculopathy grading according to the META-PM classification is presented in Figure 3. 

The number of eyes in each category of the ATN classification is shown in Table 1.

The number of eyes in each atrophic category is correlated with the META-PM classification because their criteria are identical. We obtained 29 possible combinations, resulting in a mixture of the three categories. The most frequent combinations were as follows: 78 eyes with A1T0N0 (43.57%), 27 eyes with A2T0N0 (15.08%), and 13 eyes with A3T0N2 (7.26%). According to this classification, 125 eyes (69.8%) from our sample had PM.

The results obtained when using the IMI OCT classification are shown in Table 2.

Lacquer cracks identified as linear Bruch membrane defects (BMD) on structural OCT frames were found in 18 eyes (10.11%) (Figure 4), whereas 7 eyes showed signs of active CNV at the time of the study (Figure 5), and 32 (17.88%) showed identifiable inactive CNV, the so-called Fuch’s spot (Figure 6). The mean subfoveal choroidal thickness of the eyes with CNV was 75.66 ± 38.5 µm (range, 27–133 µm). When evaluating the CNV location, 2 (33.33%) eyes were subfoveal and 4 (66.67%) perifoveal, whereas 11 (33.33%) eyes had a subfoveal Fuch’s spot, 21 (63.63%) had a perifoveal spot and 1 eye demonstrated a spot > 3 mm from the fovea. 

A BMD associated with patchy or macular atrophy was found in 42 (23.46%) eyes. A BMD associated with peripapillary atrophy was found in 83 (46.37%) eyes (Figure 7). 

A total of 45 eyes (24%) had extrafoveal retinoschisis, of which 20 (46.5%) had inner and outer retinoschisis, 11 (25.64%) had inner retinoschisis only, and 6 had purely outer retinoschisis. Two eyes had either inner or outer retinoschisis and a lamellar hole and two eyes had both inner and outer retinoschisis and a lamellar hole (Figure 8).

We also found that 28 (15.6%) eyes had a DSM. Only one eye had associated subretinal fluid and OCTA excluded the presence of CNV (Figure 9). We also found a RSM in six eyes, as defined by the IMI OCT classification, four of which were found via vertical scans and were associated with inferior staphyloma. 

We also evaluated the optic nerve head in the retinography scan and found that 70 (39.1%) eyes had a tilted optic nerve head. Myopic conus was present in 73 (40.72%) eyes; 59 eyes (80.08%) had it temporally, 10 nasally and 3 inferiorly, and 1 had a myopic conus exceeding one quadrant. More extensive peripapillary atrophy was present in 83 eyes (46.37%), and of them, 60 eyes (72.29%) had minor circumpapillary atrophy, 20 eyes (24.1%) had atrophy extending toward the macular area, and 3 eyes had atrophy extending toward the fovea. 

Four eyes had a macular pit, two of which were associated with scleral dehiscence (Figure 10 and Figure 11).

Six eyes had scleral dehiscence without a macular pit. Scleral dehiscence and macular pits were both found in areas of chorioretinal atrophy. Choroidal cavitation was detected in six eyes (Figure 12).

Seventy-seven eyes (43%) had vascular folds in the HD51 scan (Figure 13). PSV were detected in 159 eyes (88.8%) and 16 of them were related to a Fuch´s spot and 4 eyes with active CNV. Seven eyes had PSV with lacquer cracks.

Concerning the retinal, choroidal, and scleral foveal thicknesses, six eyes were excluded due to their inability to detect the fovea, mainly because of macular atrophy (Figure 14).

Only 132 eyes were evaluated to measure the FAZ and its perimeter due to segmentation errors at the superficial plexus and 127 for the deep plexus. The results from those eyes are shown in Figure 15.

Only 166 fundus retinographies could be analyzed to study the cilioretinal arteries due to problems of focus or loss of transparency that made the task difficult. We found a cilioretinal artery in 20 eyes (12.04%), and the different types of cilioretinal arteries are presented in Figure 16.

Table 3 shows a comparison among other studies of FAZ measurements. Table 4 shows a comparison among different studies of the retinal, choroidal, and scleral foveal thicknesses.

## 4. Discussion

In this study, multimodal imaging was used to detect the main complications of the eyes with PM. We used structural SS-OCT and SS-OCTA supported by retinography to detect macular lesions related to PM. To our knowledge, this is the largest study that uses a Caucasian sample and focuses on a high number of lesions, which is important, as Spanish PM cohorts may not behave in the same way as Asian cohorts [22]. Our study highlights the importance and potential of the use of widefield SS-OCT to detect less frequently reported complications in PM, such as macular pits, choroidal cavitations, scleral dehiscence, PSV, or vessel folds.

We evaluated the presence of staphyloma using HD spotlight 16 mm line scans with which we could identify and define the size, location, and limits of staphylomas in many patients, which are difficult to assess with other OCT tools whose scan size may be smaller or produce mirror artifacts that make image interpretation difficult. The percentage of patients with staphyloma is smaller compared to an early study published by our group (92.7%) [22]. This may be explained by the fact that our previous study recruited patients only from the retina unit, which may have introduced bias toward more severe pathology; this study included patients from other specialties, such as refractive surgery, and the findings were not only based on the OCT assessment performed in the present study, which also might have affected the results. In the study of Haarman et al. [38], a prevalence rate of up to 43% of posterior staphyloma was found, which was more common with older age. This number is smaller than ours, but the study of Haarman et al. was based on the general population. Our results are slightly lower than those of Shinohara et al. [39] (75% of eyes in an Asian sample), but the authors also found that the wide macular and narrow macular types were most prevalent, similar to the results of our study and to that of Frisina et al. [40], who also found wide macular and narrow macular types to be the most common types of staphyloma in a Caucasian HM sample.

A multimodal approach to detect lacquer cracks is advised, mainly because lacquer cracks are sometimes small defects that can be missed by OCT [41]; thus, we evaluated the presence of lacquer cracks by combining retinography and OCT that are viewed as linear BMDs. The percentage of lacquer cracks found in the current study is similar to our previous results [22] and slightly lower than that found by Fang et al. (14.7%) [42] in an Asian sample, whereas Park et al. [33] found higher values (24.5%); however, the AL in their sample was longer than ours, which may explain the differences.

We used OCT and OCTA to detect CNV and Fuch’s spot. However, it is difficult sometimes to distinguish between the two, since flow can also be detected by OCTA in some Fuch’s spots. The neovascular vessels also may be difficult to detect in the en-face image of OCTA due to the extreme retinal thinning in patients with PM, changes in the curvature of the posterior pole, and the presence of segmentation errors. In fact, a differential diagnosis must often be carried out considering the patient’s symptomatology. The percentage reported in this study is similar to our previous study [22]. Fang et al. [42] found higher percentages in their PM cohort of 26.9%, but a smaller value in their overall HM and PM cohorts of 17.3%. Our results also are higher than those of Park et al. [33] of 7.6%, although their mean AL was also longer. When comparing the results of atrophic maculopathy to those obtained in the previous study published by our group [22], we also found that category 1 was the most frequently found category. The main difference arises in category 3, as our previous study found a higher percentage for this category, which may be explained again by the differences in the recruitment origin that have been previously mentioned. Our results differed significantly from those of Ruão et al. [23], in that they found a prevalence for patchy atrophy (category 3 in our study) of 79.9% compared to our value of 16.8%. They also found higher values for CNV, which was present in 61.9% of their eyes, since their recruitment was carried out in the macula unit of a large hospital, which also may have introduced bias toward the presence of more severe symptomatic pathology. A comparison of the results obtained in our study with large population studies with different designs may not be valuable [19].

We found that 69.8% of our sample had MM, as defined by the META-PM classification, which differs from that obtained by Haarman et al. [38], who found a prevalence of 25.9% in a Dutch HM sample. The differences may be due to the different ethnicities and the fact that our sample had a longer AL; their recruitment process also differed in that the study of Haarman et al. was population-based.

The use of the ATN classification [4] can help us to better characterize all pathological features that can be found in patients with PM. It is interesting that none of the five most common combined categories had a tractional component of T1 or higher, which is understandable considering that in our sample, 89.4% of eyes did not have any tractional component. We also must consider that the T component refers to tractional damage to the macular area, but does not consider extramacular tractional damage and because of this, these extrafoveal features are missed in the ATN classification. Another limitation of this classification is that the neovascular dominion includes lacquer cracks, active CNV, and Fuch´s spots, but one patient may have more than one; however, only one of them may be detected. Nevertheless, this classification provides a very useful understanding of PM. The use of the IMI OCT classification [3] enables a complete evaluation of the patients with HM and PM without using any tools except OCT, and it is the only classification that includes DSM/RSM in the definitions of PM. OCT is the only tool that facilitates the diagnosis of most of the complications of HM. We believe that any patient with HM should undergo this examination regularly. Finally, as the definitions and classifications have changed throughout the past 20 years, it is difficult to compare results between different studies.

In our study, we looked for extrafoveal retinoschisis, in addition to the foveal retinoschisis already assessed with the ATN classification. In our evaluations, we found that inner plus outer retinoschisis was the most frequent presentation. Similar results were found in the study of Ruão et al. [23], who found extrafoveal retinoschisis in 23.9% of their patients’ eyes. However, Xiao et al. reported that the most prevalent type of extrafoveal retinoschisis in their sample was the inner type [43], although they selected a sample of patients with a previous diagnosis of extrafoveal retinoschisis, which may explain the difference.

The prevalence of DSM in our sample was within the range previously published [44]. Interestingly, only one eye had subretinal fluid due to this condition, although it is the main recognized complication of DSM, whose prevalence has been reported to vary from 2% to 67% [44]. We also observed two eyes with RSM that was unrelated to inferior staphylomas, which was hypothesized to be an early sign of evolving DSM that is more frequently found in younger patients with HM [44].

Alterations in the optic nerve were common in our sample. Nearly half of the patients had circumpapillary atrophy, which extended toward the temporal side and involved the macular area in 23 eyes, whereas 40% of patients had a myopic crescent, unlike the results published by Haarman et al. [38], who reported a value of up to 80% of peripapillary atrophy in their sample, with a higher frequency in older patients, although they do not distinguish between myopic conus and circumpapillary atrophy. He et al. [45] identified peripapillary atrophy in 112 of 134 eyes and this was the most frequent sign observed in HM, whereas Fang et al. [42] reported a prevalence of 89.5%. All these figures along with our results indicate the high prevalence of this kind of lesion, which has already been reported. The evaluation of this sign is important because it is the most frequent sign of progression of PM and this area of peripapillary atrophy can extend toward the macula, as the previously mentioned authors found.

Macular pits, defined as focal excavations in atrophic areas [26,46,47], are viewed as focal depressions in the OCT scans, usually in areas of complete chorioretinal atrophy. In the fundus images, a greyish lesion inside the atrophic area is indicative of the presence of the pit. These lesions can be defined as hypofluorescence in fundus reflectance images of OCT (Figure 10). To our knowledge, no prevalence rates have been reported because only a small number of case series have been published. We found four eyes with a macular pit and two of them were accompanied by scleral dehiscence. In addition, six eyes had scleral dehiscence without a macular pit. All of them were found in atrophic areas. The cause of this kind of lesion remains unclear, although the uneven force of intraocular pressure in those areas of a debilitated ocular wall may affect its appearance [26].

Freund [48] first described ICC as pigmented retinal epithelium detachment in the peripapillary area. Toranzo et al. [49] later reported no pigmented retinal epithelium detachment and suggested its current name. ICC can be observed in structural OCT as a hyporeflectant space where the choroid should be. The frequency with which we observed this anomaly is similar to the one obtained by Shimada [50]***m*** who reported a 4.6% prevalence rate in their sample. You et al. [51] reported a higher prevalence in the Beijing Eye Study in 15 of 89 subjects with HM with OCT images that showed ICC. In addition, Haarman et al. [38] reported similar prevalence levels of ICC in a Caucasian sample. These numbers indicated that ICC is not an uncommon alteration and wide-field SS-OCT is key to correctly detect and characterize it.

Forty-three percent of the patients’ eyes had vessel folds, which is indicative of the rigidity of the vessel in contrast to the prevalent sign of retinal elongation in our study. Sayanagi et al. [52] also reported vascular folds in an HM sample, but that case series included only seven eyes.

We also identified PEV in 88.8% of eyes, a sign that some authors believe is a possible connection to CNV [53,54]. In fact, we did find an association between PEV and lacquer cracks, CNV, and Fuch´s spots in some eyes, but most eyes did not have any of these conditions or the PEVs were not in the same location. Since this finding is so frequent, it may also well be that colocalization of PEV and CNV occurs by chance.

We could not analyze the FAZ in all of the eyes due to segmentation errors, which made measurement difficult. When comparing our results with normal subjects, an increase in SPFAZ was observed across studies [55,56,57,58,59]. DPFAZ shows greater heterogeneity, with some studies showing smaller values [56,58,59], while others reported higher values [55,57]. These results indicate a change in the FAZ area of myopes compared to healthy subjects, which is probably caused by the elongation of the ocular layers. However, these comparisons should be considered carefully because the equipment and methodology vary greatly across studies. Table 3 shows an overview of the measurements of FAZ in different studies. The SPFAZ values are equal to the results published by Xiuyan et al. [30], although their sample consisted of moderate myopes, but the values were smaller than those reported by Lee et al. [31]. However, the DPFAZ area was slightly larger in our study than the one published by Lee et al. [31], but smaller than that reported by Xiuyan et al. [30].

We manually measured the retinal, choroidal, and scleral foveal thicknesses using the HD spotlight vertical line scan. Although seven eyes had to be eliminated because it was impossible to identify the fovea, we measured the fovea in most of the studied eyes. Our results indicate high dispersion, which may be explained by the different alterations that occur in myopic eyes, such as the different types of staphylomas, CNVs, DSMs, or the presence of macular atrophy that can affect the measurements. Global elongation results in thinning of the ocular layers [33,34,35,36,60]. This is evident when comparing the thicknesses of the HM eyes to those of healthy subjects [37]. Our results showed some variations compared to the published data. We found thinner mean retinal thicknesses compared to healthy adults, as expected [37]. However, our retinas were thicker compared to the results reported by Liu et al. [32] and Maruko et al. [35], but thinner than those reported by Xiuyan et al. [30]. Thus, it is possible to observe heterogeneity in the values obtained in the scientific literature. The results for choroidal and scleral thicknesses also showed thinner values compared to healthy eyes, with values that are nearly half of those published [37,61]. If we focused on studies about HM samples, high heterogeneity can also be detected. For choroidal thickness, Park et al. [33], Wong et al. [34], and Maruko et al. [35] reported smaller values than those obtained in our study. In contrast, Tan et al. [37] and Xiuyan et al. [30] reported higher values compared with the current study. Again, these differences may be explained by the heterogeneity of PM with different types of macular alterations that may affect the choroidal thickness. These differences also apply to the scleral thickness in that our results were higher than those of Park et al. [33], Wong et al. [34], Maruko et al. [35], and Hayashi et al. [36]. However, differences of up to 70 µm can be observed across studies that may be based on measurement methodologies, sample characteristics, or the type of OCT used.

We found that 12.5% of eyes had cilioretinal arteries in our sample, which is slightly lower than that reported by Zhu et al. [62], who found that 17.05% of their patients’ eyes had a cilioretinal artery or the 14.5% reported by Meng [29]. The most common type of cilioretinal artery in the current study was a temporal ribbon, which is consistent with the results of Meng et al. [29] who used the same classification. The importance of this cilioretinal artery arises from the study of Zhu et al. [62], in that the results suggested that the presence of a cilioretinal artery may increase the macular vascular flow, resulting in better VA, which we could not verify with our data.

HM and PM are important causes of visual loss, but the existing literature on the associated causative pathologies is insufficient. Furthermore, studies have been conducted primarily in Asian countries. Thus, our results contribute to a better understanding of the frequency of observations of macular complications of HM and PM in a Caucasian sample. Likewise, it highlights the importance of novel equipment, such as SS-OCT, in the diagnosis of this pathology, as the PLEX^®^ Elite 9000 OCT was the main diagnostic tool used in this study.

The main limitations of this study were the sample characteristics. We recruited patients with HM who sought consultation in an ophthalmic clinic. Thus, the frequency of observations that has been presented may be biased toward more severe pathology, in that patients with high myopes without symptoms may not seek ophthalmologic consultation. Nevertheless, we included the healthiest patients from the refractive unit to minimize this bias. Another limitation arises from the maximal length of the OCT spotlight scan used to diagnose staphyloma. Although 16 mm is long, it may be insufficient to detect all kinds of staphylomas, and optic nerve staphylomas may be more difficult to detect.

The current study was large and demonstrated the use of this tool to better assess the posterior pole in most patients with HM. We encourage clinicians to perform wide-field OCT in all patients with HM.

## 5. Conclusions

In this study, we presented the frequency of observations of lesions in the posterior pole using OCT in a HM Spanish Caucasian cohort, which may be important for epidemiologic purposes and for a better understanding of PM. The role of wide-field OCT as the main diagnostic tool to detect macular alterations in PM has been highlighted. SS-OCT is a novel tool that is more potent than previous generations of OCTs and can detect most major complications of PM in the posterior pole, such as retinoschisis or CNV, and less frequent alterations, such as macular pits or ICC. Most importantly, some pathologies, such as perforating scleral vessels, can only be identified with this new equipment; these vessels seem to more common than previously thought and they seem to be unrelated to CNV.

## Figures and Tables

**Figure 1 jcm-12-01846-f001:**
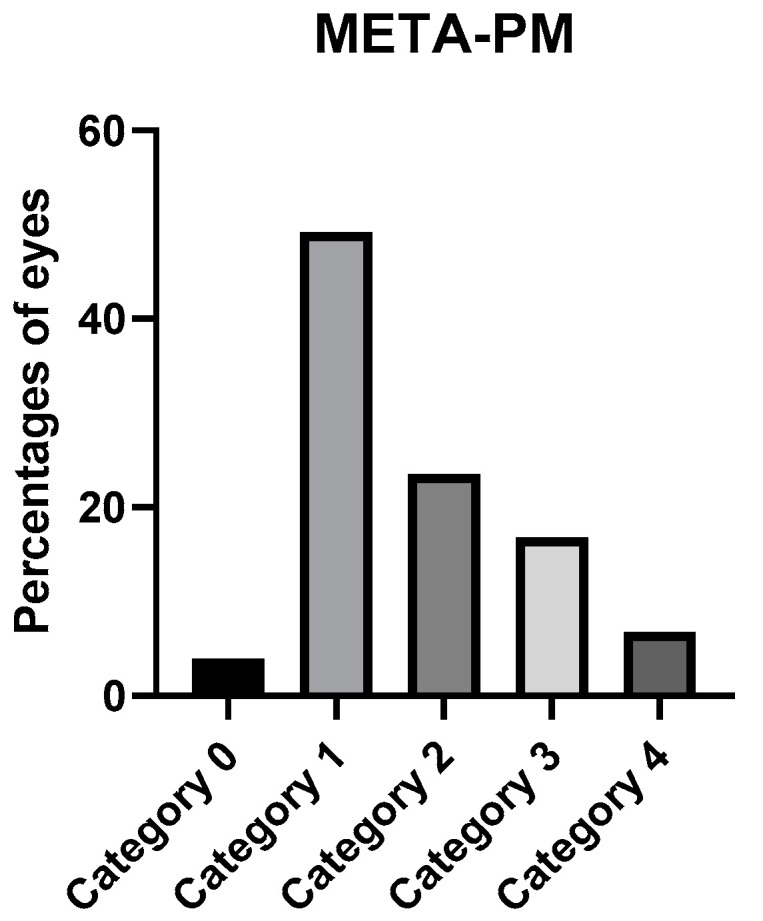
Types of detected staphylomas.

**Figure 2 jcm-12-01846-f002:**
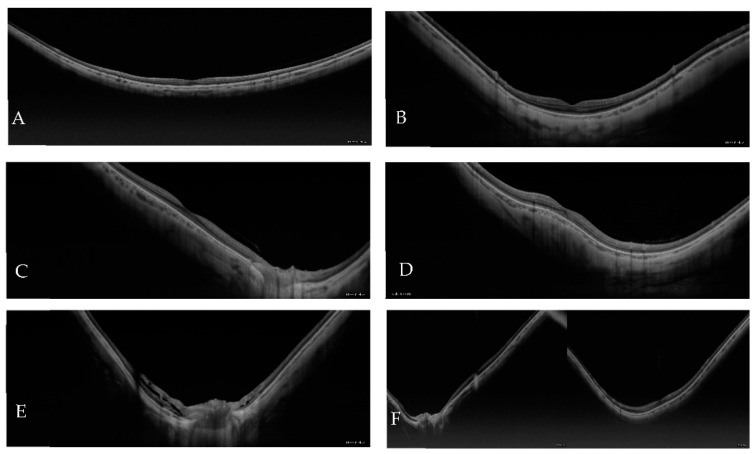
Types of staphylomas: (**A**) wide macular; (**B**) narrow macular; (**C**) nasal; (**D**) inferior; (**E**) peripapillary, and (**F**) other types. All images are horizontal in orientation except D.

**Figure 3 jcm-12-01846-f003:**
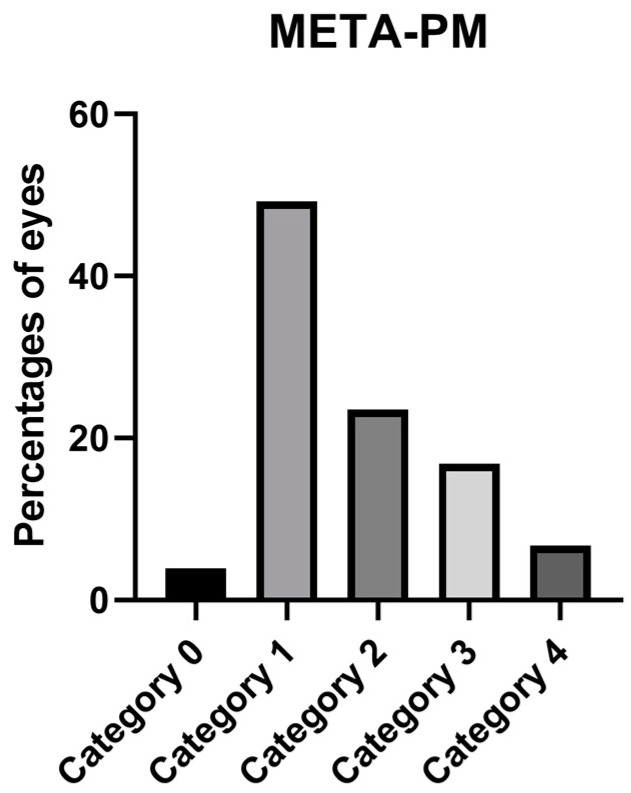
Percentage of eyes in the categories of the META-PM classification.

**Figure 4 jcm-12-01846-f004:**
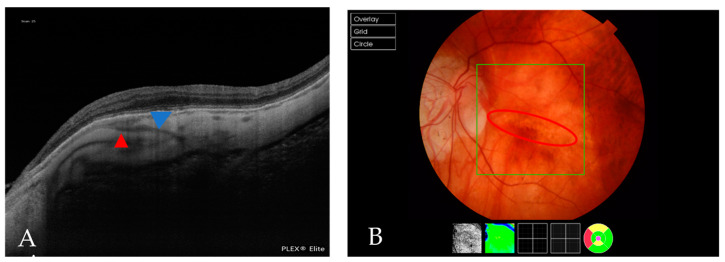
(**A**) An OCT scan that shows a BMD caused by a lacquer crack (blue arrowhead). The red arrowhead indicates a PEV reaching the choroid. (**B**) A fundus image of the same patient, in which the red circle indicates a lacquer crack.

**Figure 5 jcm-12-01846-f005:**
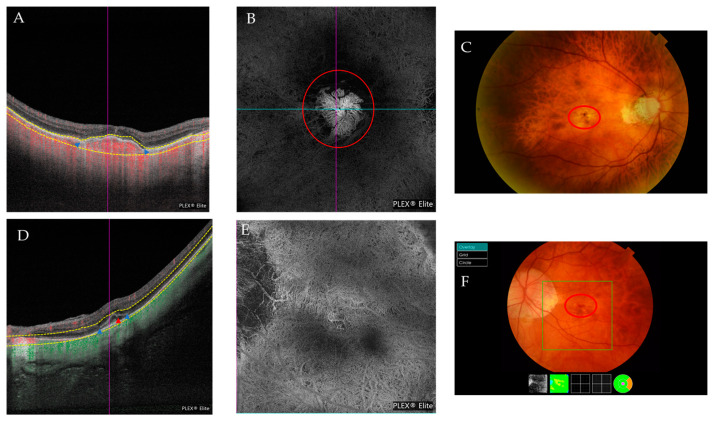
(**A**–**C**) Multimodal imaging from the same patient with a Fuch’s spot. (**A**) The blue arrowheads indicate the limits of the CNV. (**B**) The outer retina to the choriocapillaris (ORCC) on the OCTA images with a tailing artifact removed from the en-face image. The red circle indicates the CNV. (**C**) The red circle indicates the position of the CNV obtained via retinography. (**D**,**E**) Multimodal imaging from the same patient presenting with active CNV. (**D**) An OCT scan that shows signs of CNV activity with retinal thickening due to subretinal fluid and small intraretinal cysts (red arrowhead). Its limits are indicated by the blue arrowheads. (**E**) En-face ORCC from an OCTA image of the same patient. (**F**) The red circle indicates the presence of the membrane and blood in this patient’s retinography.

**Figure 6 jcm-12-01846-f006:**
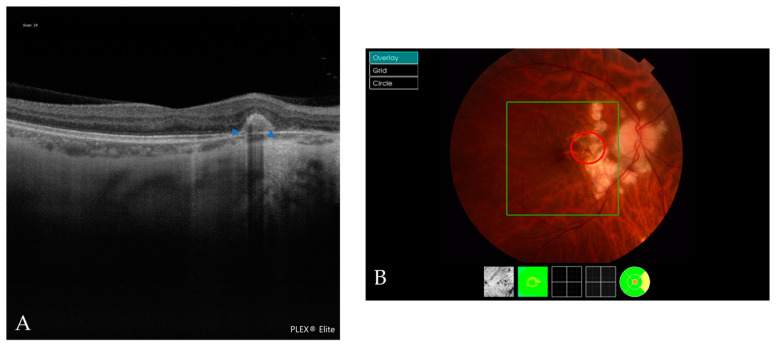
Multimodal image of a Fuch´s spot. (**A**) An OCT scan shows a Fuch´s spot (blue arrowheads). (**B**) The red circle indicates the Fuch´s spot in the retinography.

**Figure 7 jcm-12-01846-f007:**
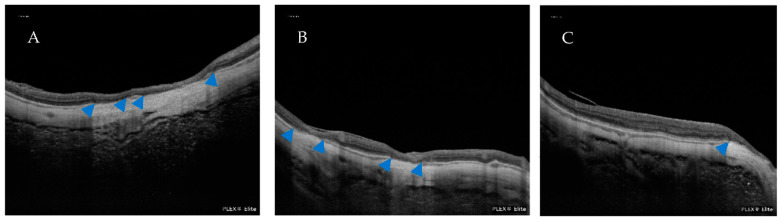
(**A**) An OCT scan of the BMDs associated with patchy atrophy. (**B**) An OCT scan of a BMD involving the fovea. (**C**) An OCT scan of a BMD associated with peripapillary atrophy. Blue arrowheads indicate the presence of BMDs.

**Figure 8 jcm-12-01846-f008:**
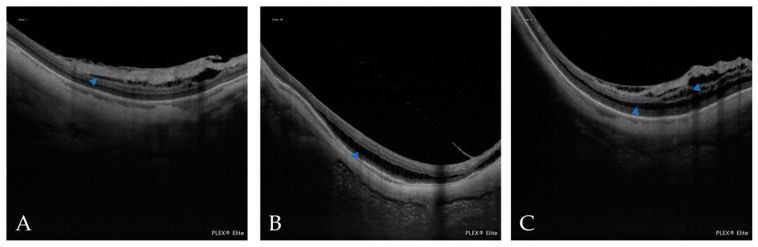
(**A**) An OCT scan of inner retinoschisis. (**B**) An OCT scan of outer retinoschisis. (**C**) An OCT scan of inner and outer retinoschisis. Blue arrowheads indicate the presence of retinoschisis.

**Figure 9 jcm-12-01846-f009:**
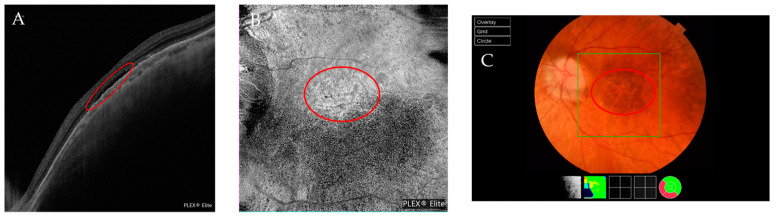
Multimodal images of a DSM. (**A**) In an OCT scan, the red circle indicates the presence of subretinal fluid. (**B**) In an en-face image, the red circle indicates a hyperreflective area correlated with the subretinal fluid. (**C**) In a retinography image, the red circle indicates the area where subretinal fluid is present.

**Figure 10 jcm-12-01846-f010:**
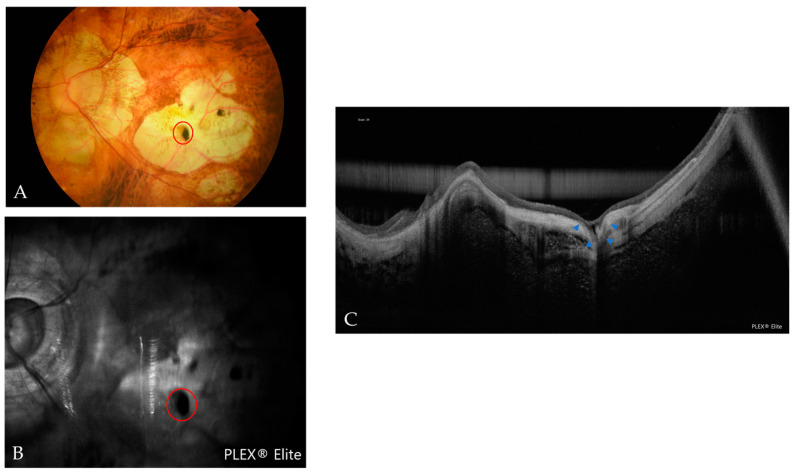
Multimodal images of a macular pit with scleral dehiscence. (**A**) In a fundus photograph, the red circle indicates a grayish and black lesion that is correlated with the pit. (**B**) In an OCT reflectance image, the red circle indicates an area of hyporeflectance correlated with the pit. (**C**) In an OCT scan, the blue arrowheads indicate the pit with scleral dehiscence.

**Figure 11 jcm-12-01846-f011:**
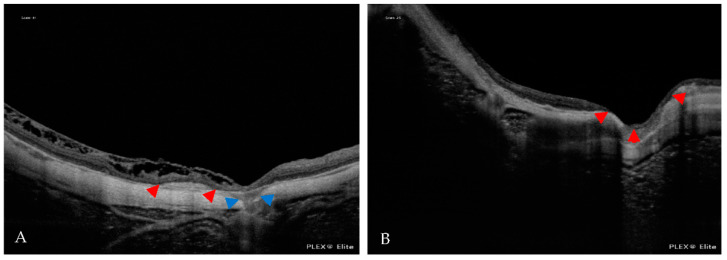
(**A**) In an OCT scan, the blue arrowheads indicate scleral dehiscence. The red arrowheads indicate a Fuch´s spot. (**B**) In an OCT scan, the red arrowheads indicate the macular pit.

**Figure 12 jcm-12-01846-f012:**
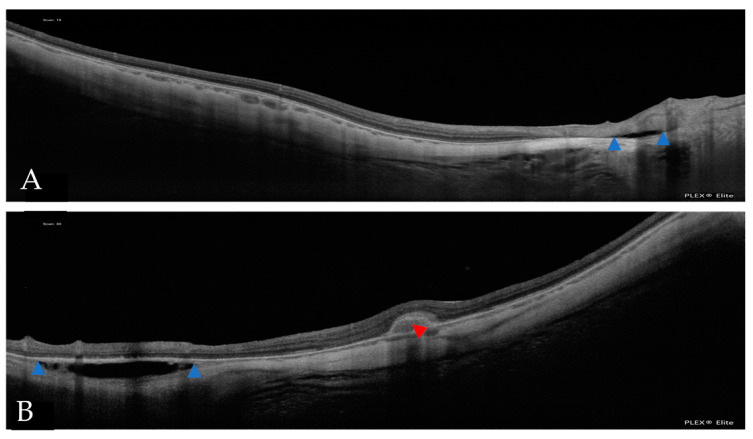
In the OCT scans, the blue arrowheads indicate intrachoroidal cavitation (ICC). (**A**) A small ICC is near the optic nerve head. (**B**) A large ICC is in the vicinity of the CNV (red arrowhead).

**Figure 13 jcm-12-01846-f013:**
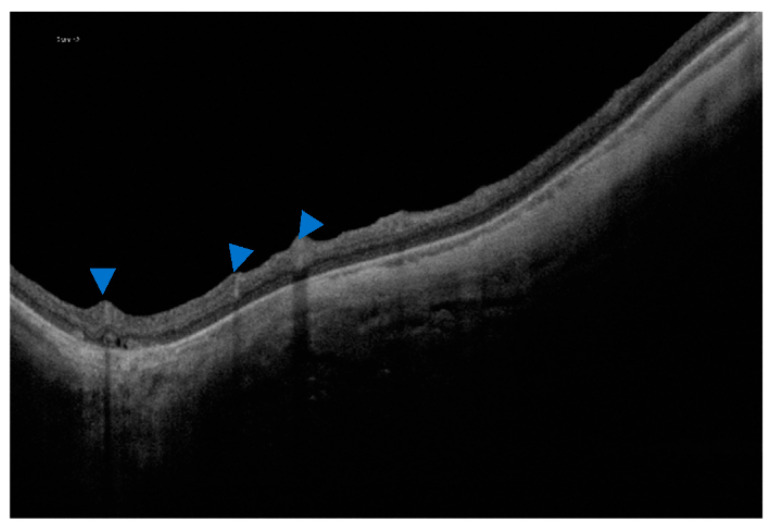
In an OCT scan, the blue arrowheads indicate vascular folds.

**Figure 14 jcm-12-01846-f014:**
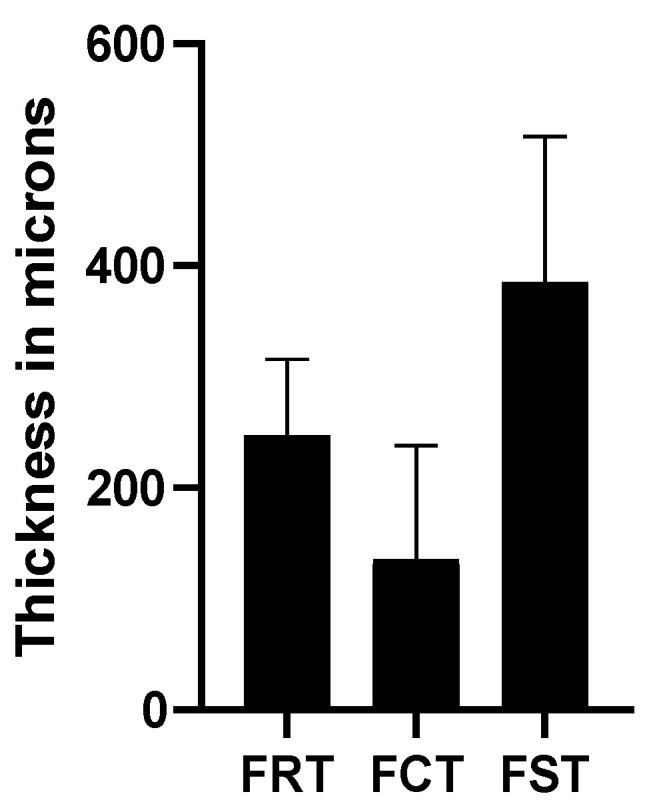
Retina, choroidal, and scleral foveal thicknesses. FRT = foveal retinal thickness; FCT = foveal choroidal thickness; FST = foveal scleral thickness.

**Figure 15 jcm-12-01846-f015:**
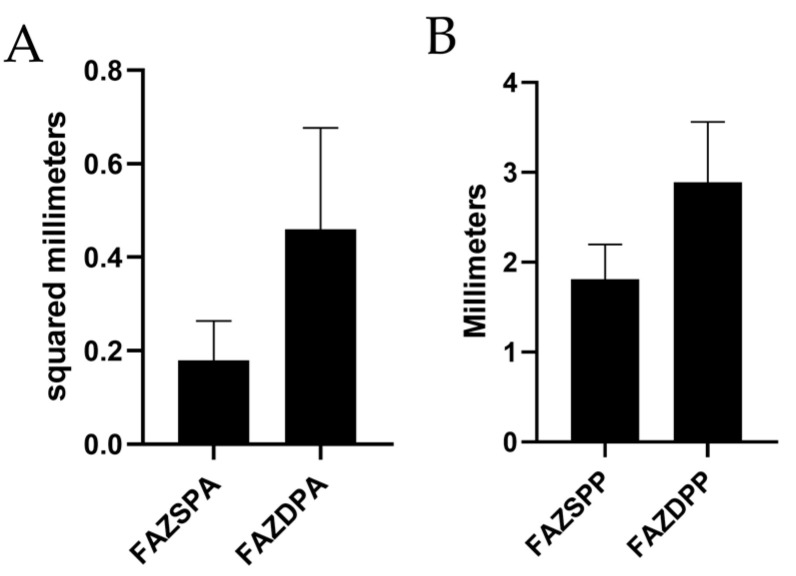
The mean area (**A**) and perimeter (**B**) for superficial and deep plexuses in the FAZ. FAZSPA = foveal avascular zone and superficial plexus area; FAZSPP = foveal avascular zone and superficial plexus perimeter; FAZDPA = foveal avascular zone and deep plexus area; FAZDPP = foveal avascular zone and deep plexus perimeter.

**Figure 16 jcm-12-01846-f016:**
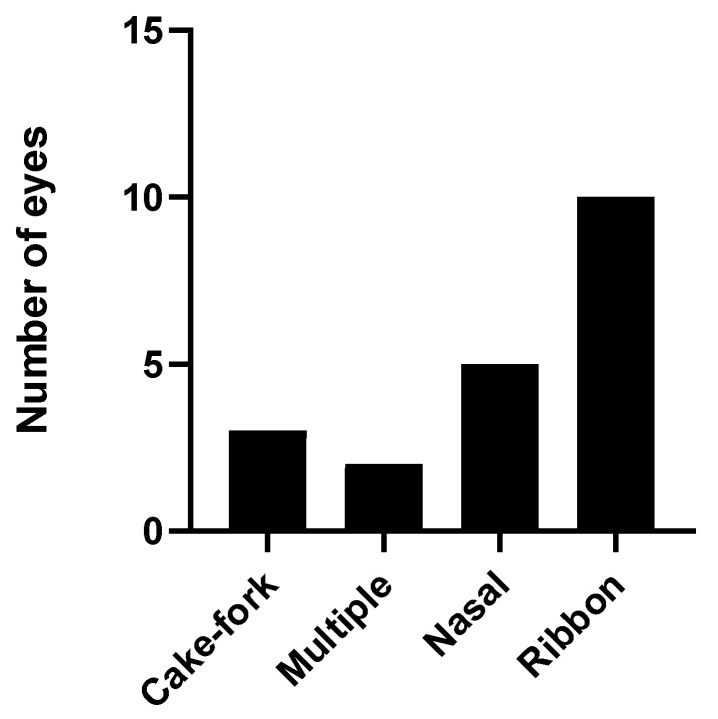
Number of eyes with each type of cilioretinal artery.

**Table 1 jcm-12-01846-t001:** Number of Eyes in Each ATN Category.

Atrophic Classification	No. of Eyes	Tractional Classification	No. of Eyes	Neovascular Classification	No. of Eyes
A0	7	T0	160	N0	136
A1	87	T1	6	N1	12
A2	41	T2	9	N2a	7
A3	32	T3	2	N2s	27
A4	12	T4	2		

**Table 2 jcm-12-01846-t002:** Number of Eyes in Each Category of the IMI OCT Classification.

IMI OCT Classification	No. of Eyes
No MM	88
Ia	19
Ib	27
II	12
III	5
IIIa	17
IIIb	11

**Table 3 jcm-12-01846-t003:** Comparison of the FAZ Measurements among Different Studies.

**Authors and Countries**	**Xiuyan et al. [30]** **China**	**Lee et al.** [31] **Republic of Korea**	**Arlanzón et al.** **Spain (Present Study)**
**No. of eyes**	154	30	179
**Baseline age (years) mean ± SD**	21.80 ± 1.32	55.5 ± 13.5	50.58 ± 14.98
**Mean spherical equivalent ± SD**	−4.06 ± 2.26 D	−9.98 ± 5.03 D	−6.74 ± 6.44D
**SPFAZA (mm)**	0.1835 ± 0.0477	0.37 ± 0.15	0.18 ± 0.084
**DPFAZA (mm)**	0.3299 ± 0.0601	0.43 ± 0.16	0.46 ± 0.217

SPFAZA = superficial plexus foveal avascular zone area; DPFAZA = deep plexus foveal avascular zone area.

**Table 4 jcm-12-01846-t004:** Comparison of Retinal, Choroidal and Scleral Layer Thicknesses among Different Studies.

**Authors and Countries**	**Xiuyan et al.** [30] **China**	**Liu et al.** [32] **China**	**Park et al.** [33] **Republic of Korea**	**Wong et al.** [34] **Singapore**	**Maruko et al.** [35] **Japan**	**Hayashi et al.** [36] **Japan**	**Tan et al.** [37] **China**	**Arlanzón et al.** **Spain (Present Study)**
**No. of eyes**	154	30	237	92	58	75	38	179
**Baseline age (years) mean ± SD**	21.80 ± 1.32	24.43 ± 3.43	63.0 ± 11.6	60.2 ± 8.4	65.5 ± NA	62.3 ± 11.3	NA	50.58 ± 14.98
**Mean spherical equivalent ± SD in diopter**	−4.06 ± 2.26	−7.85 ± 1.37	−15.4 ± 5.4	−12.5 ± 5.1	−12.8 ± 3.6	−12.9 ± 4.1	−7.35 ± 1.1	−6.74 ± 6.44
**Retinal thickness (μm)**	252.14 ± 17.33	240.91 ± 13.36	NA	NA	206 ± 92	NA	NA	247.06 ± 68.87
**Choroidal thickness (μm)**	232.16 ± 56.65	NA	29.2 ± 21.7 with staphyloma 46.9 ± 39.3 without staphyloma	82.0 ± 57.12 in mild MMD 31.5 ± 0.5 in severe MMD	52 ± 38	NA	253.8 ± 71.0	135.95 ± 102.44
**Scleral thickness (μm)**	NA	NA	268.7 ± 95.9 with staphyloma 316.2 ± 76.5 without staphyloma	297.0 ± 73.8 in mild MMD 261.6 ± 78.5 in severe MMD	335 ± 130	284.0 ± 70.4	NA	385.40 ± 131.41

MMD = myopic macular degeneration; NA = not applicable; SD = standard deviation.

## Data Availability

Authors can make their research data available, accessible, and usable if necessary.

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
