# Peer review of "Does PLEX® Elite 9000 OCT Identify and Characterize Most Posterior Pole Lesions in Highly Myopic Patients?"

_jcm, 2023, doi:10.3390/jcm12051846_

Round 1
Reviewer 1 Report
The authors present an interesting report on the importance of wide-field OCT in detecting posterior pole lesions in pathological myopia.
The study is well conducted and the manuscript is well written.
Here are a few comments to improve the manuscript:
1. Methods: page 3 line 112: " Those included a 6x6 OCTA scan centered on the macula and a 12x12 HD51 sc... 6x6 HD51 was performed instead." Kindly mention how many were 6x6 and how many were 12x12
2. Methods: Page 3 line 118 "In addition, the next lesions on ... other macular lesions." The definitions of the individual lesions need to be improved.
3. Figure 15 shows both area and length measurements, while the y-axis is only in millimeters. Either a separate graph for area and perimeter can be used, or the values can be presented in the text.
4. Results: page 11 line 317: "FAZ" instead of "FAV"
5. Minor changes are required in grammar and syntax throughout the manuscript.
Author Response
REVIEWER 1 EVALUATION
The authors present an interesting report on the importance of wide-field OCT in detecting posterior pole lesions in pathological myopia. The study is well conducted, and the manuscript is well written. Here are a few comments to improve the manuscript:
- Methods: page 3 line 112: " Those included a 6x6 OCTA scan centered on the macula and a 12x12 HD51 sc... 6x6 HD51 was performed instead." Kindly mention how many were 6x6 and how many were 12x12.
Answer: we have incorporated the exact number of 6x6 and 12x12 scans made in lines 113-114.
- Methods: Page 3 line 118 "In addition, the next lesions on ... other macular lesions." The definitions of the individual lesions need to be improved.
Answer: A more thorough definition of the individual lesions has been implemented and added some references. The definitions extend from lines 120 to 144.
- Figure 15 shows both area and length measurements, while the y-axis is only in millimeters. Either a separate graph for area and perimeter can be used, or the values can be presented in the text.
Answer: figure 15 has been changed. A new figure with a separate graph for area and perimeter has been created. Additional information has been implemented in Figure 15 description line 326.
- Results: page 11 line 317: "FAZ" instead of "FAV"
Answer: this minor vocabulary typo has been corrected and changed to FAZ in line 336.
- Minor changes are required in grammar and syntax throughout the manuscript.
Answer: as mentioned above, the manuscript and its grammar and syntax were revised by a native English speaker.
Reviewer 2 Report
This study is interesting and is worth to be published. However, some remarks will make it more valuable:
There is a lot of abbreviations, and it is often hard to follow the results especially that authors refers to several classifications.
OCT B-scans pattern is not well defined.
ATN Category are not defined in methods.
Table 3 would be better discussed in discussion section.
It's better to define Fuch's spot in methods to avoid any confusion in results and in discussion.
Author Response
REVIEWER 2 EVALUATION
This study is interesting and is worth to be published. However, some remarks will make it more valuable: There is a lot of abbreviations, and it is often hard to follow the results especially that authors refer to several classifications.
- OCT B-scans pattern is not well defined.
Answer: additional information of the number of HD51 6x6 and 12x12 has been incorporated. Furthermore, the type of OCT scans is already stated, and other technical information can be consulted in Plex Elite 9000 manual guide.
- ATN Category are not defined in methods.
Answer: ATN Category is well defined in the referred paper and its criterion has been followed to classify the pathology. Please consult: Ruiz-Medrano J, Montero JA, Flores-Moreno I, Arias L, García-Layana A, Ruiz-Moreno JM. Myopic maculopa-thy: Current status and proposal for a new classification and grading system (ATN). Prog Retin Eye Res. 2019; 69:80-115.
- Table 3 would be better discussed in discussion section.
Answer: table 3 results are discussed in section discussion lines 497-502.
- It's better to define Fuch's spot in methods to avoid any confusion in results and in discussion.
Answer: a more thorough definition of Fuch´s spot has been implemented in lines 121 to 123.